# Profiling crRNA architectures for enhanced Cas12 biosensing
Elizabeth Toyin Ajibode ⓘ , Alexandra R. Bender & Kevin Yehl ⓘ ✉

CRISPR-Cas diagnostics are revolutionizing point-of-care molecular testing due to the programmability, simplicity, and sensitivity of Cas systems with *trans*-cleavage activity. CRISPR-Cas12 assays are promising for detecting single nucleotide polymorphisms (SNPs). However, reports vary widely describing Cas12 SNP sensitivity, and an underlying mechanism is lacking. We systematically varied crRNA length and valency to investigate the role of crRNA architectures on Cas12 biosensing in the context of speed-of-detection, sensitivity, and selectivity. Our results demonstrate that crRNAs complementary to 20 base pairs of the target DNA is optimal for rapid and sensitive detection, while a complementary length of 15 base pairs is ideal for robust SNP detection. Additionally, we uncovered a unique periodicity in SNP sensitivity based on nucleotide position and developed a structural model explaining what drives Cas12 SNP sensitivity. Lastly, we showed that bivalent CRISPR-Cas sensors have synergistic and enhanced activity that is distance dependent.

CRISPR-Cas diagnostics have garnered significant interest due to the discovery of CRISPR-Cas systems with *trans*-cleavage activity, specifically Cas12, Cas13, and their subtypes[1,2]. Like Cas9, this class of effectors are RNA guided nucleases that can be programmed to cut nucleic acid sequences proximal to a protospacer adjacent motif (PAM), which is a short nucleic acid sequence generally comprising 3–6 nucleotides respective to the Cas[3,4]. However, unlike Cas9, Cas12 and Cas13 remain active and nonspecifically cleave nearby ssDNA or RNA, respectively[4]. This '*trans*-cleavage' activity, also referred to as 'collateral-cleavage', and has been harnessed for nucleic acid biosensing through the addition of nucleic acid reporter molecules to the reaction buffer, whereby reporter hydrolysis results in detectable signal[5]. This signal amplification enables sensitive dsDNA, ssDNA, and RNA detection at the point-of-care without requiring sophisticated instrumentation[6–8]. Therefore, significant efforts are being devoted to understanding parameters important to assay performance and optimizing accordingly.

Towards this goal, Huyke et al. systematically investigated Cas12- and Cas13- enzyme kinetics and how it affects assay sensitivity[9]. This work was motivated by the underlying principle that the limit-of-detection (LoD) of Cas12- and Cas13-based assays are inherently governed by enzyme kinetics, but these rates were poorly understood and inconsistently reported. Huyke et al. characterized several Cas12 and Cas13 types with many different crRNAs (14 crRNAs) and quantified the *trans*-cleavage catalytic efficiency to be between $10^5 – 10^6\,M^{-1}S^{-1}$[9]. They also concluded that low background activity governs the LoD, which is picomolar when using a fluorescent substrate.

We aim to expand upon these efforts by investigating how crRNA architectures, specifically length and valency, affect Cas12 biosensing regarding enzyme kinetics and assay -sensitivity and -selectivity[10]. We hypothesize that crRNA molecules with greater complementarity to a target nucleic acid sequence will have improved sensitivity, but will come as a trade-off for distinguishing homologous sequences. The rationale being crRNAs with greater complementarity to target DNA exhibit larger ΔG of binding but will also have a reduced ΔΔG between a perfect match sequence and sequences containing mismatches. Figure 1a summarizes this hypothesis and illustrates the anticipated trade-off. Furthermore, this work investigates factors important for detecting single-nucleotide polymorphisms (SNP) using Cas12, as SNPs are the root cause to many diseases[11–13]. However, similar to Cas12 kinetic studies, little is known about what drives Cas12 sensitivity to SNPs, and prior reports are vary widely in describing Cas12 SNP sensitivity[14]. Herein, we propose a structural model that potentially explains Cas12 sensitivity to SNPs.

Lastly, we investigate whether Cas12 multivalency can be used as a general strategy for improving sensitivity and selectivity of CRISPR-Cas biosensors (Fig. 1b), but more broadly, to all CRISPR-Cas systems, as multivalent systems can exhibit cooperative binding. This results in a slower $k_{off}$ and thus a smaller $K_d$. Also, cooperative binding produces a sharper binding isotherm, which allows for better differentiation between perfect and mismatch sequences (Fig. 1c). Together, cooperativity has the potential to increase the percentage of activated Cas12 at lower target DNA concentrations, thus improving sensitivity, while simultaneously improving selectivity.

## Results

### Optimizing conditions to saturate cas12-crRNA complex formation and DNA binding

Cas12 *trans*-cleavage activity is dependent upon the formation of a ternary complex: Cas12 assembles with crRNA, and the ribonucleoprotein-complex

Department of Chemistry and Biochemistry, Miami University, Oxford, Oxford, OH, USA. ✉e-mail: yehlk@miamioh.edu

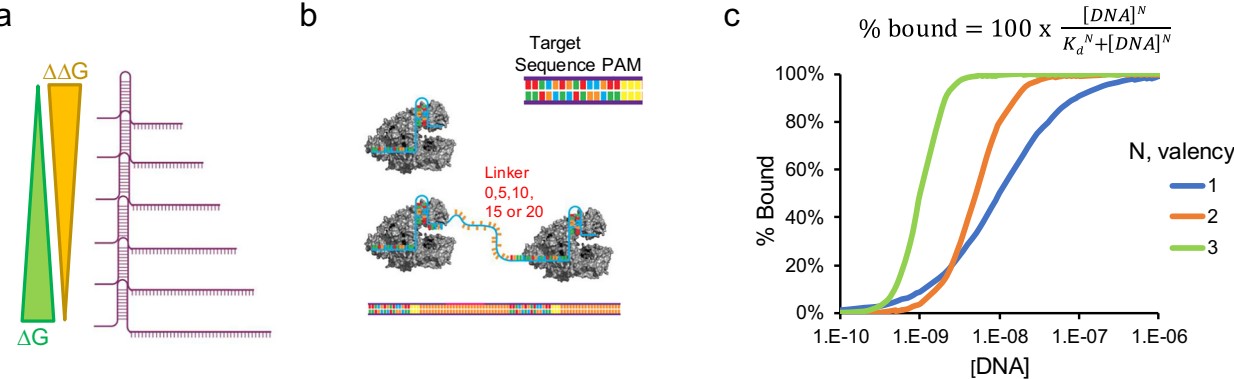

**Fig. 1 | Proposed model for how crRNA influences Cas12 binding to target DNA. a** Schematic illustrating the relationship between crRNA length and binding to target DNA. **b** Proposed arrayed crRNA structures for assembling multivalent Cas-biosensors. **c** A plot showing hypothetical binding for cooperative multivalent systems.

binds to target DNA, which activates *trans*-cleavage (Fig. 2a)[15–17]. In order to accurately measure kinetic parameters, it is essential that Cas12 saturates binding to activator DNA so that the concentration for activated Cas12 can be determined, which upon saturation, is the DNA concentration. Conditions that do not satisfy saturation of DNA binding, will lead to over-estimation of [E-S] and an underestimation of kinetic parameters.

Therefore, Cas12 *trans*-cleavage activity was measured as a function of Cas12 concentration, while keeping crRNA and target DNA concentrations constant at 12.5 nM and 1 nM, respectively, and finding at what Cas12 concentration *trans*-cleavage activity becomes saturated. This concentration equates to saturation in forming the crRNA-Cas12 complex. Target DNA was set to 1 nM because we assumed this would be the upper limit for detection for a typical assay and that higher DNA concentrations would not be representative of real-world samples. Additionally, the $K_D$ for the Cas12-crRNA ribonucleoprotein complex binding target DNA has been measured to be between 0.27 and 5.0 nM[18], so conditions that saturate crRNA-Cas12 complex formation using 12.5 nM crRNA would equate to 71–98% bound target DNA. Higher concentrations of crRNA-Cas12 and DNA would increase the dynamic range of the assay, but this will come as a tradeoff for cost.

crRNA-40 was selected for initially optimizing assay conditions because we anticipated that a longer crRNA would form a weaker complex with Cas12, thus requiring more Cas12 for saturating formation of the crRNA-complex and that this concentration would be sufficient for the shorter crRNAs. We also tested crRNA-20 to confirm this hypothesis. Results are summarized in Fig. 2b, d and show that saturation in activity for crRNA-20 and crRNA-40 occur at ~20 nM and ~100 nM Cas12, respectively. Since activated Cas12 is dependent upon DNA binding, the data was fit to a $K_D$ fit model (Fig. 2c, e). The $K_D$ was measured to be 4.1 nM for crRNA-20, which agrees well with work by Nguyen et al., where they measured the $K_D$ to be ~5 nM[10]. Interestingly, the fit for the crRNA-40 showed a Hill coefficient of 4.6 and a $K_D$ of ~62 nM. We speculate that the large Hill coefficient is due to a two-step equilibrium comprising Cas12 binding both crRNA and DNA, and the $K_D$'s are similar[19,20]. Based on these results, a ~100x and ~12.5x excess of Cas12 and crRNA to 1 nM activator DNA, respectively, are required for saturation in Cas12 activity.

## Cas12 dependence on crRNA length in relation to kinetics

Next, we investigated how crRNA length affects Cas12's reaction velocity, as the reaction velocity determines the time required to achieve detectable signal. Additionally, we investigated how crRNA length affects the LoD. We hypothesized that greater complementarity between crRNA and target DNA would result in a lower LoD and better sensitivity. To test this hypothesis, *trans*-cleavage kinetics were measured for crRNAs varying in complementarity to target DNA, spanning 15, 20, 25, 30, 35 and 40 nucleotides, where crRNAs are referred to by the number of nucleotides complementary to the target DNA (i.e., crRNA-15 refers to 15

complementary sequences to the target DNA). crRNAs were transcribed from a double stranded DNA template, purified, and characterized using gel electrophoresis (Fig. 3a; Supplementary Fig. 3a). Surprisingly, crRNA-30 showed a larger band compared to the other crRNAs, which was approximately double than expected (Fig. 3a). crRNA-30's total length is 53 ribonucleotides. We speculate that this was due to dimerization of crRNA-30. Therefore, we carried out a denaturing gel and observed the 'dimerized' band go away, and that crRNA-30 followed the general trend in increasing size (Supplementary Fig. 3b).

The *trans*-cleavage kinetics were measured for each crRNA under pseudo-first order reaction conditions with respect to activator DNA (200 nM LbCas12, 12.5 nM crRNA, 1 nM activator DNA, and 100 nM reporter substrate, respectively). The initial slope was measured from the fluorescence kinetic plots and used as a proxy for reaction velocity. The time period for measuring the slopes varied depending on the rate of reaction, but generally time periods resulting in an $R^2$ greater than 0.95 were used. Our findings show that the crRNA length had to be greater than 15 nucleotide complementarity to activator DNA to induce rapid *trans*-cleavage activity, though crRNA-15 did have detectable activity (Fig. 3b–d). Surprisingly, crRNA-20 resulted in the fastest reaction velocity, where longer crRNAs decreased in activity. This differs from what Nguyen et al. observed, where longer crRNAs resulted in 3.5x increased activity compared to crRNA with a spacer length of 20 (i.e., crRNA-20). crRNA-30 had significantly lower activity compared to the observed linear trend (Fig. 3c, blue bars), which this decrease in activity corroborates the previous gel findings that crRNA-30 potentially is dimerized. It is important to note that the standard deviation is calculated from an $n = 2$, which slightly underestimates standard deviation, but the general trend of crRNA-20 > crRNA-25 > crRNA-35 > crRNA-40 > crRNA-30 > crRNA-15) holds true for each DNA concentration (Fig. 3d). Data was also compiled for crRNA-20 from two batches of Cas12 ($n = 4$), shown in Supplementary Fig. 4, which shows high reproducibility.

Since background activity has a major influence on LoD, the background activity for each crRNA was measured. The background activity largely remained the same for each crRNA, where crRNA-20 had the lowest background activity, followed by crRNA-30, -35, -40, -25, and -15. The signal enhancement for each crRNA was calculated by taking the ratio between measured reaction velocity (1 nM activator DNA) and background signal (no activator DNA), which is summarized in Fig. 3c (red stars) and plotted on a secondary axis. Standard curves of reaction velocity as a function of activator DNA were generated in order to derive the LoD for each crRNA (Fig. 3d and Supplementary Fig. 5). Specifically, the standard error was calculated from a linear regression analysis, which was divided by the coefficient of the X-variable and multiplied by 3.3. This analysis showed that crRNAs -20, -25, -35, and -40 were the most sensitive resulting in LoD values ranging between 20 and 40 pM, followed by crRNA-15 and 30, with LoD values of ~70 pM.

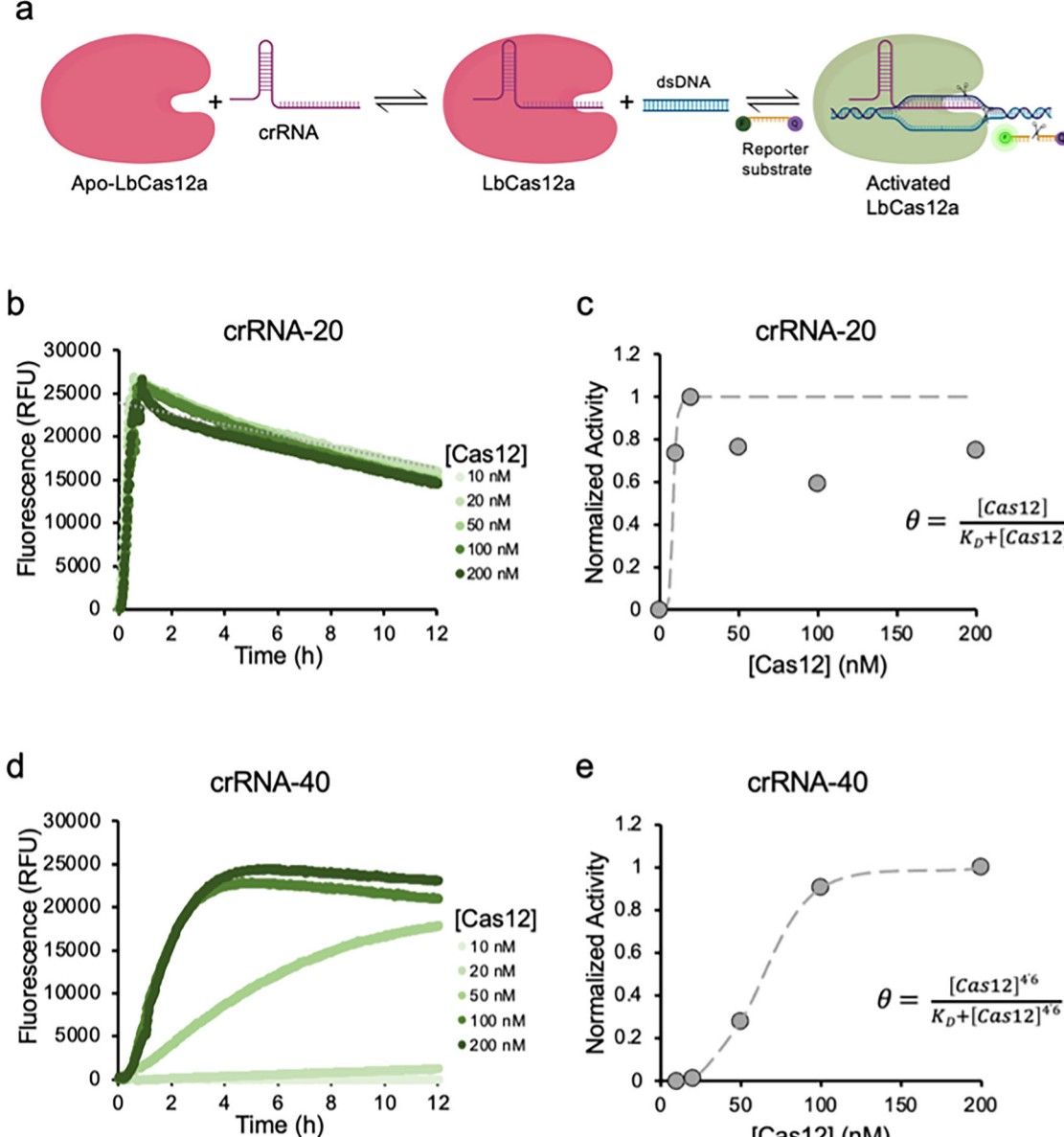

**Fig. 2 | Condition optimization for saturating target DNA binding by Cas12. a** Schematic illustrating the dynamic equilibrium for forming the activated Cas12 ternary complex. **b, d** A plot summarizing Cas12 kinetics having varying Cas12 concentration and keeping crRNA (crRNA-20 & 40) and target DNA concentrations constant at 12.5 nM and 1 nM, respectively. **c, e** A plot summarizing normalized Cas12 *trans*-cleavage activity from (**b, d**). The data was fit to a $K_d$ fit model (dashed gray line).

To better understand why crRNA-20 had the fastest reaction velocity, Michaelis-Menten kinetics for crRNA-15, crRNA-20 and crRNA-40 were compared to see how max velocity ($V_{max}$) and $K_m$ are influenced by crRNA length (Fig. 4a–f). The results show that for crRNA-15, both $K_m$ and $V_{max}$ are significantly compromised. We believe that crRNA-15 does saturate DNA binding, indicated by linear response to varying DNA concentration (Fig. 3d), but is unable to induce a conformational change required for activating significant *trans*-cleavage activity. Whereas, for crRNA-40, the main contribution for decreased activity is from reduced Cas12 *trans*-cleavage turnover ($k_{cat}$), though $K_m$ is not insignificant. We believe crRNA-40 has slight reduced binding to reporter DNA by ~30%, determined by comparing $K_m$'s between crRNA-20 and crRNA-40. We believe this is due to increased ionic repulsion from the additional nucleotides in the crRNA and reporter substrate. However, the max velocity varied significantly between the two crRNAs, differing by approximately 4-fold, where crRNA-20 had the fastest $V_{max}$. Together, these results show that Cas12 complexed

with different length crRNAs have varying $K_m$'s for reporter substrate but significantly differ in enzymatic turnover ($k_{cat}$).

Next, to simulate detection similar to a real-world setting, we tested the ability of crRNA-20 to detect target DNA in a mixed pool of non-target DNA using T7 genomic DNA as the mixed pool DNA and varying up to 10x excess by mass. Results are summarized in Supplementary Fig. 6, where sensitivity is unaffected. Surprisingly, we observed a general trend for increased sensitivity. This corroborates seminal works by Chen et al., Nalefski et al., and Smith et al., which show Cas12 preferentially targets ssDNA and and polyT for *trans*-cleavage over dsDNA respectively[5,21,22].

### Cas12 dependence on crRNA length in relation to detecting single-nucleotide-polymorphisms

Since SNP mutations are responsible for many diseases and are important for genotyping cancers, we set out to measure Cas12 SNP sensitivity as a function of crRNA length[23,24]. We hypothesized that shorter crRNA

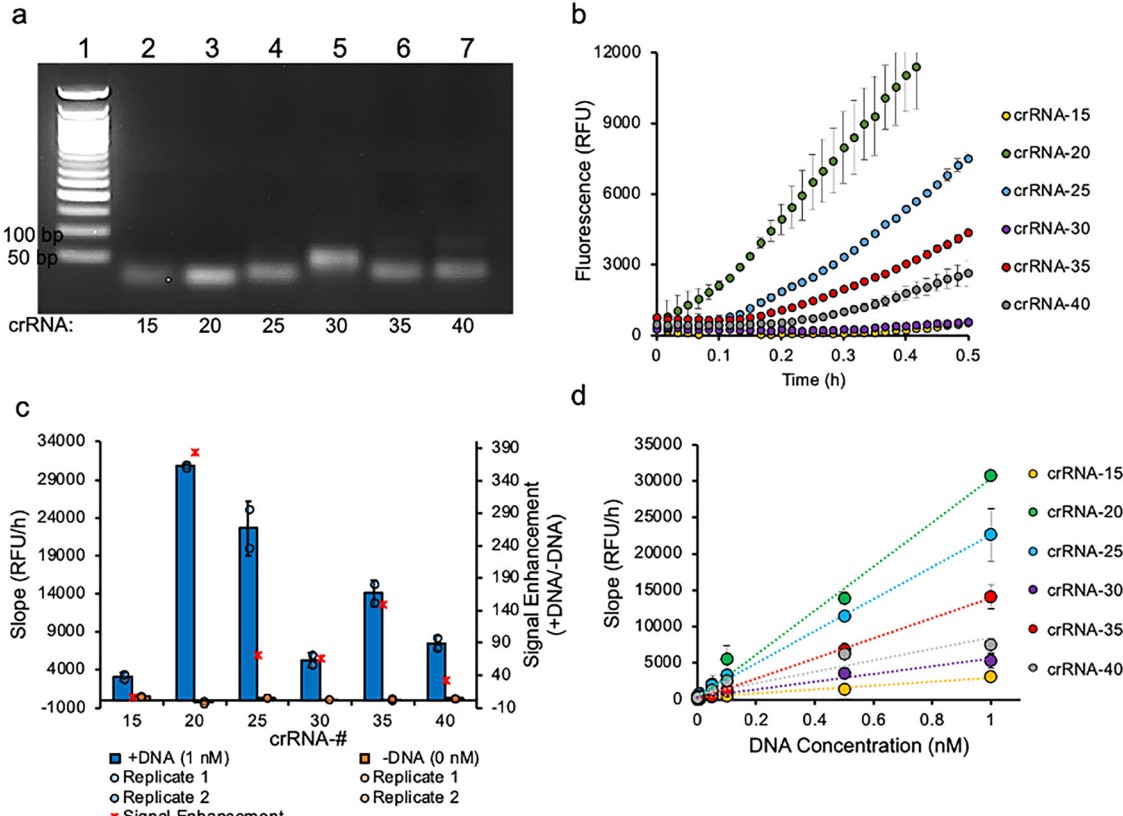

**Fig. 3 | Cas12 kinetics with monovalent crRNA varying in length. a** Gel electrophoretic characterization of the transcribed crRNAs. Lane 1 is the DNA ladder and lanes 2–7 are the different crRNAs starting from crRNA-15 to crRNA-40, respectively. **b** A summary of Cas12a *trans*-cleavage kinetics with different lengths crRNA at 1 nM target DNA concentration. **c** Comparison of the average Cas12 *trans*-cleavage activity for different length crRNAs with (blue bars) and without (orange bars) activator DNA, which was calculated from the slope of the linear portion of the curve in (**b**). The dot-plot represents individual data points from independent experiments. The signal enhancement (SE, red stars) is plotted on a secondary axis and is calculated from dividing Cas12 activity in the presence of activator by Cas12 background activity in the absence of activator. **d** Standard curves for each crRNA showing average Cas12 *trans*-cleavage activity for varying concentrations of activator DNA. Error bars show standard deviation ($n$ = 2).

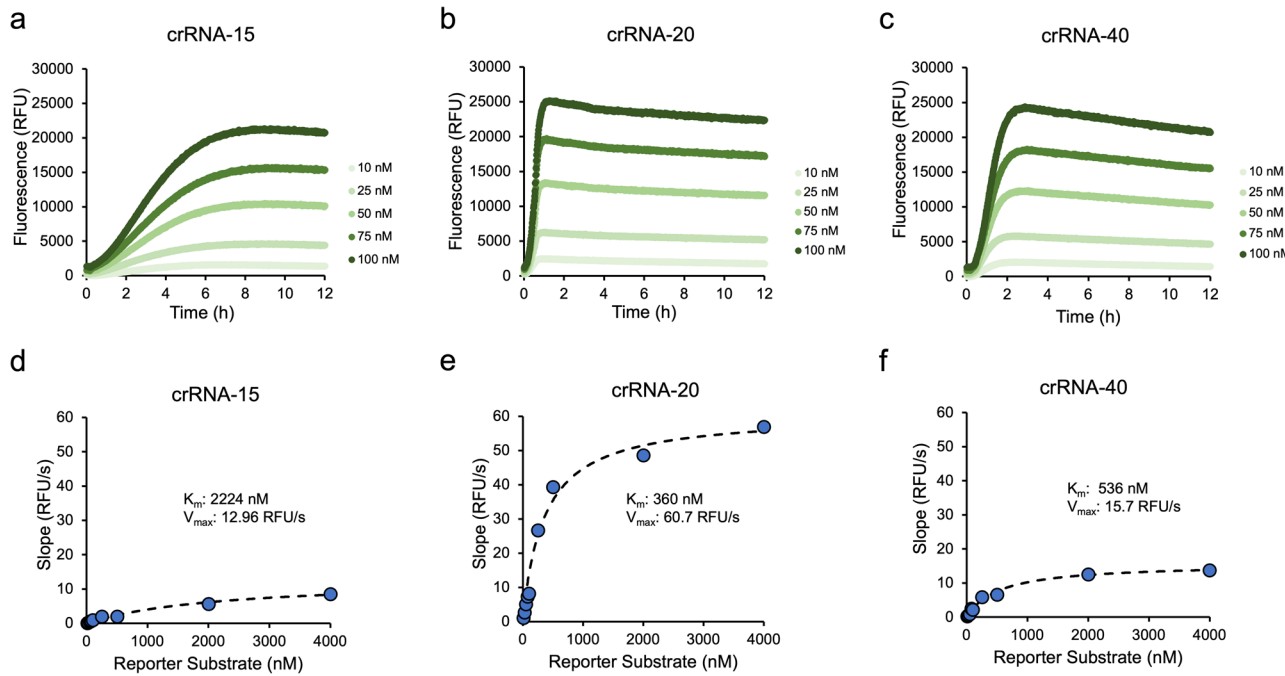

**Fig. 4 | Michaelis-Menten kinetics for Cas12 *trans*-cleavage activity for crRNA-15, -20 and -40.** A summary of Cas12 *trans*-cleavage kinetics for crRNA-15 (**a**) crRNA-20 (**b**) and crRNA-40 (**c**) with varying substrate concentration. The concentrations of Cas12, crRNA and DNA were 50 nM, 12.5 nM, and 1 nM respectively. **d–f** Michaelis-Menten plots generated from (**a–c**), respectively. The inset shows the fitted Michaelis-Menten parameters resulting in the best fit ($K_m$ = 2224 nM, 360 nM and 536 nM for crRNA-15, crRNA-20 and crRNA-40, respectively).

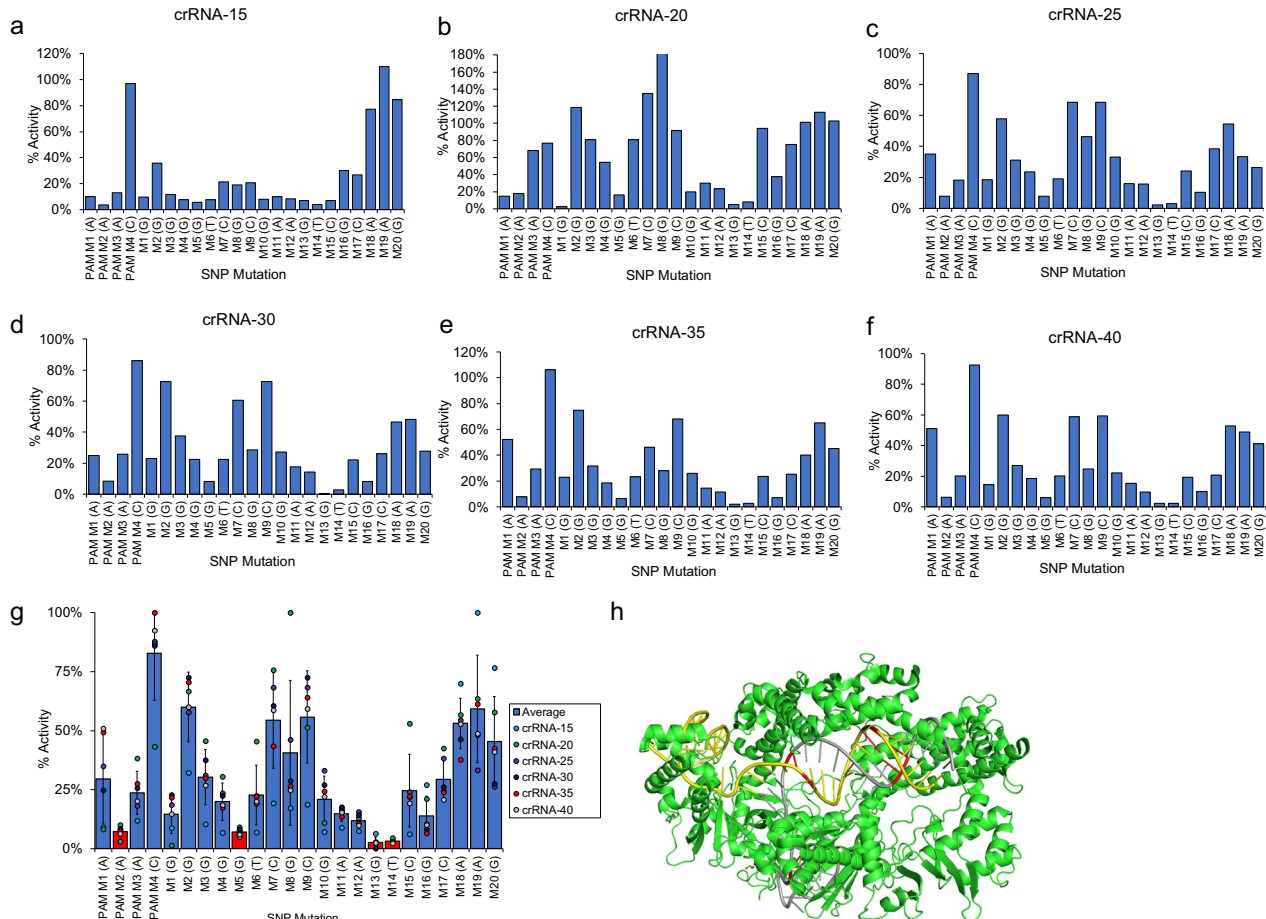

**Fig. 5 | Cas12 SNP selectivity studies.** A summary of Cas12 SNP sensitivity for each crRNA (**a–f**). Activities are normalized to WT activity. **g** A plot showing the average normalized activity for each nucleotide position across all crRNAs. The dot-plot represents individual data points. Error bars show standard deviation ($n = 6$). The red bars indicate nucleotide positions within the crRNA that were most sensitive to

SNP mutations having activities <10% compared to the most active nucleotide position. **h** A crystal structure showing LbCas12a bound to crRNA and DNA (PDB:5XUZ), where Cas12 is colored green, dsDNA is colored yellow, crRNA is colored gray, and sensitive nucleotide positions in both crRNA and DNA are colored red.

sequences would result in a larger difference in *trans*-cleavage activity between perfect match and mismatch sequences, since shorter crRNAs have a larger $\Delta\Delta G$ between perfect match and mismatch sequences. Cas12 *trans*-cleavage activity was measured for linear activator DNA containing SNP mutations across the PAM and up to 20 nucleotides downstream, where the 20th position refers to the 3′ terminus of the target strand (Supplementary Fig. 7). SNP mutations were made by exchanging a purine for a pyrimidine at each position along the target strand or vice versa. Cas12 *trans*-cleavage activity for each target DNA sequence are summarized in Fig. 5a–g and was normalized to the perfect match sequence for that respective crRNA.

The results show that crRNA-15 had the highest sensitivity to SNP mutations compared to all other crRNAs, where M1 through M15 had significantly diminished activity, less than 10% compared to WT. The longer crRNAs showed less of a difference in activity compared to WT, but interestingly, there seemed to be a periodicity to the SNP sensitivity that repeated approximately every 6-7 nucleotides (Fig. 5b–f). To better observe this trend, the averaged normalized activity for each position was plotted and is shown in Fig. 5g and Supplementary Fig. 8. Positions that had the highest SNP sensitivity were positions PAM M2, M5, M13, and M14. We hypothesized that this trend was due to the helical structure of the DNA-RNA duplex and how Cas12 binds to target DNA, in a 'sandwiched' structure. A 6-7 nucleotide periodicity corresponds to a half turn in the helix, suggesting that two opposite sides of the helix are most sensitive, presumably the sides sandwiched by Cas12. To test this model, we analyzed the crystal

structure of LbCas12a complexed with crRNA and activator dsDNA (Fig. 5h), where the nucleotide positions most sensitive to SNP mutations are colored in red. These are nucleotide positions that had <10% of normalized activity. Interestingly, the sensitive nucleotides had the largest steric interaction with Cas12, and were positioned at either side of the helix, lending support to the proposed model.

**Investigation into bivalent crRNA for improving Cas12 biosensing**

To investigate whether Cas12 biosensing can further be improved by crRNA architecture, we designed bivalent crRNA. We anticipated that bivalent crRNA would cooperatively bind to target DNA, and this would result in a stronger binding affinity compared to monovalent crRNA. We also anticipated that cooperative binding would lead to greater selectivity, and that there would be an optimal linker spacing between tandem crRNAs dependent upon the distance between the two binding sites in the target DNA (Fig. 1b). Additionally, there is most likely a minimal crRNA linker length required to accommodate multiple Cas effectors onto a single crRNA. Since Cas12 is known to process its own crRNA at the 5′ terminus, a hetero-bivalent crRNA was designed, comprising a Cas12a crRNA and a Cas9 sgRNA in the 5′ to 3′ orientation, respectively (Supplementary Fig. 1b)[25].

To systematically investigate the role of linker length on Cas12 activity, bivalent crRNAs of crRNA-20 were transcribed and purified that varied in linker lengths by five ribonucleotide increments ranging between 0 and 20 ribonucleotides. The resulting bivalent crRNAs were characterized through

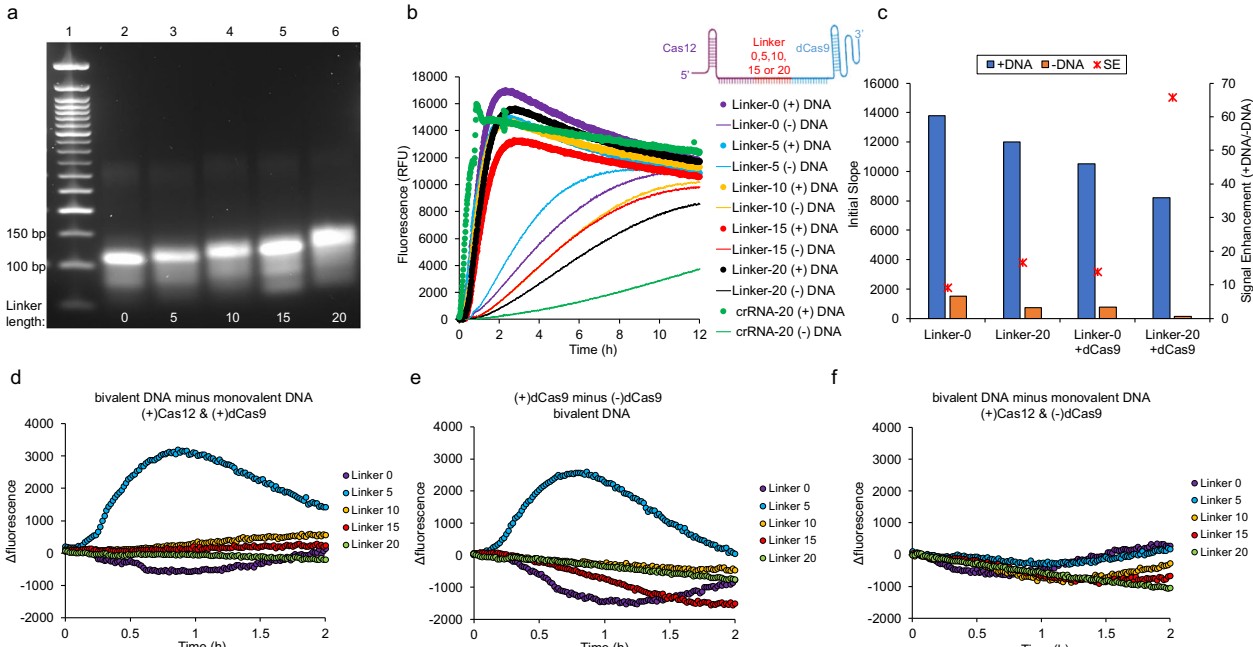

**Fig. 6 | Cas12 activity with bivalent crRNA. a** Gel electrophoresis characterization of bivalent crRNAs. Lane 1 represents the 50 bp DNA ladder, lanes 2–6 represent the different length of bivalent crRNA in the order 142, 147, 152, 157, and 162 ribo-nucleotides respectively. The gel shows a progressive upward shift as the spacer length of bivalent crRNA increases. **b** Overall activity of Cas12 with all bivalent crRNAs without dCas9. **c** Comparison of Cas12 *trans*-cleavage activity for crRNAs having different spacer lengths with (blue bars) and without (orange bars) activator DNA and in the presence and absence of dCas9. Signal enhancement is shown in red stars and plotted on a secondary axis. **d** Cas12 activation in the presence of dCas9 with bivalent and monovalent DNA. **e** Cas12 activation in the presence and absence of dCas9 with bivalent DNA only. **f** Cas12 activation in the absence of dCas9 with monovalent and bivalent DNA.

gel electrophoresis and are referred to by their linker length (Linker-##) (Fig. 6a). Cas12 activation was carried out using linear target DNA, where the corresponding Cas binding sites are separated by 20 base pairs. To ensure that linear activator DNA did not alter Cas12 activation compared to plasmid DNA, comparative studies were carried out between the two, and no differences were observed (Supplementary Fig. 9). The reaction conditions were similar to the monovalent studies, aside changing the concentration of Cas12 and Cas9 to 100 nM each and 25 nM for crRNA. Cas12 *trans*-cleavage kinetics were initially measured in the absence of dCas9, which all bivalent crRNAs had similar activities comparable to monovalent crRNA (Fig. 6b, darker colored curves). However, background activity was significantly higher compared to monovalent crRNA (Fig. 6b, lighter colored curves). To confirm that the high background activity was not due to DNase contamination or template contamination from the in vitro transcription reaction, we tested cleavage activity of the reporter DNA substrate in the absence of target DNA by comparing Cas12 only (without crRNA) vs. crRNA only vs. crRNA + Cas12 for crRNAs: crRNA-20, Linker-0, and Linker-15 (Supplementary Fig. 10). We also tested activation of crRNA-20 + Cas12 with varying concentrations of template DNA used for the in vitro transcription reaction (Supplementary Fig. 11). Together, the results show that we do not have DNase contamination, nor does template DNA activate Cas12 to the same level, though we do have high background activity for the bivalent crRNAs.

Regardless, we continued to explore how the addition of dCas9 altered binding and subsequent Cas12 activation to activator DNA containing two corresponding binding sites. The data are summarized in Fig. 6d–f for each bivalent crRNA varying in linker length, where we compared fluorescence by taking the difference between (i) bivalent DNA vs monovalent DNA having both Cas12 and Cas9; (ii) bivalent DNA with Cas12 and Cas9 vs without Cas9; and (iii) bivalent DNA vs monovalent DNA with Cas12 and no Cas9. Remarkably, the addition of dCas9 drastically reduced background activity, while only slightly decreasing overall activity in the presence of activator DNA. This resulted in a signal enhancement of ~65x when comparing Linker-20 in the presence of dCas9 with and without target

DNA, where only a signal enhancement of ~8x was observed in the absence of dCas9 comparing with and without target DNA (Fig. 6c).

Results also show synergistic activity for Linker-5. Only in the presence of a bivalent target DNA, Cas12, and Cas9 is enhanced activity observed by comparing the difference in fluorescence (Fig. 6d, e). Comparing bivalent versus monovalent DNA without Cas9 shows no enhancement (Fig. 6f). These results show that bivalency enhances activity but is distance dependence. The linker distance requirement is approximately 5 RNA nucleotides to span the 20 base pair distance between the Cas12 and Cas9 binding sites.

## Discussion

In this study, we systematically investigated how crRNA architectures alter Cas12 kinetics in the context of improving speed-of-detection, sensitivity, and selectivity for point-of-care CRISPR-based diagnostics. We found that crRNA with 20 nucleotide complementarity to target DNA led to the fastest reaction velocity. Additional investigations revealed that the decrease in reaction velocity for longer crRNAs was mostly due to a decrease in $V_{max}$ with little contribution due to weaker binding to reporter substrate, as one would have expected due to the additional negative charge. Both crRNA-20 and crRNA-40 had similar Michaelis-Menten constants of 360 nM and 536 nM, respectively. These values agree with the upper range of what Huyke et al. measured[9]. This reduced turnover could potentially be due to induced conformational change in Cas12, similar to how Cas9 is sometimes inactivated by extended crRNA.

As anticipated, shorter crRNA (crRNA-15) enabled better SNP discrimination, where SNP mutations between 1 and 15 positions can be detected by comparing to WT activity and observing greater than 90% difference in activity. Previously, SNP detection could only reliably be achieved by measuring Michaelis-Menten kinetics and comparing $K_m$, which is laborious and requires ~ten measurements per sample assuming five measurements are needed to generate a Michaelis–Menten curve and comparing samples to a WT standard[14]. Surprisingly, longer crRNAs had a periodicity to SNP sensitivity, where nucleotide positions most sensitive to SNP mutations were spaced 6–7 nucleotides apart. Through analysis of the

crystal structure, these positions are the ones in closest contact to the Cas12 protein, sandwiched on opposite sides of the helix. This analysis provides an insight into the relationship between structure and selectivity in detection, which potentially explains the discrepancies described in prior reports, as different Cas12 subtypes have different structures. Furthermore, different types of SNP mutations may result in less subtle structural perturbations compared to the ones tested herein, thus not revealing the trend.

Lastly, we investigated how Cas multivalency improves biosensing. Hetero-bivalent Cas complexes were synthesized using a long tandem crRNA template comprised of Cas12 crRNA and Cas9 sgRNA and varying liker length. Even though long crRNAs resulted in an increase in background activity for Cas12, when coupled with dCas9, background activity was significantly suppressed, which improved signal enhancement by ~65x (Fig. 6c). We also observed synergistic Cas12 *trans*-cleavage activity for hetero-bivalent crRNA having a 5 ribonucleotide linker and activator DNA having two binding sites separated by 20 base pairs only in the presence of dCas9.

It will be important to determine the generalizability of this approach. We believe that shorter crRNAs will always be ideal for detecting SNPs, but determining the optimal length currently requires empirical testing. A generalizable model linking thermodynamic parameters and induced conformational changes in Cas12 with Cas12 *trans*-cleavage kinetics will be valuable for readily designing biosensors for emerging threats. Such a model can help explain differences observed for longer crRNAs, where we observed decreased activity and Nguyen et al. observed enhanced activity[10]. Furthermore, it will be important to systematically investigate the distance dependence for multivalent CRISPR biosensors, as distance between CRISPR-Cas binding sites within a gene target is specific to the gene. Though, great efforts have been made to engineer Cas9 variants to recognize an array of PAMs[26]. Lastly, bivalent CRISPR-Cas biosensors offer unique programmability because the binding module (dCas9) is decoupled from the biosensing module (Cas12). This offers a novel approach for biosensor design, with hopes of unlocking cooperativity to rationally improve biosensor function[27].

Bi-functional Cas complexes had previously been investigated, but not in the context of biosensing and cooperativity. Hao et al. engineered a bivalent dCas9 for controlling DNA looping, and thus gene expression[24]. Their system fused strong heterodimerizing leucine zippers to the C-termini of orthogonal dCas9s, *Streptococcus pyogenes* (*Sp*) and *Streptococcus thermophilus* (*St*), and they showed that looping can be achieved over a 12 kb distance[28,29]. A complicated kinetic model was required to determine the optimal Cas concentrations required to prevent the 'Hook Effect'[28]. Wu et al. engineered a covalent dCas9-dCas12a protein fusion for folding DNA nanostructures. This approach circumvented the complicated kinetics of self-assembly and the Hook effect, but it is not readily programmable to synergistically bind sites having varied distances, as the system described here, as protein engineering is required[30].

## Methods
### Materials and equipment
All oligonucleotides were purchased from Integrated DNA Technologies (IDT) and are summarized in Supplementary Table 1. *Lachnospiraceae* bacterium Cas12a (LbCas12a, cpf1) was purchased from New England Biolabs. KAPA2G Robust Hot start PCR kit (product #KK5517) was ordered from Kapa Biosystems. All fluorescence measurements were acquired using a BioTek Synergy H1 microplate reader. Fluorescent gel images were recorded using a ChemiDoc MP Imaging System (Bio-Rad Laboratories) using a SYBR Safe DNA gel stain.

### In vitro transcription and purification of crRNA
All crRNA were made through in vitro transcription of a double-stranded DNA (dsDNA) template encoding the crRNA (Supplementary Fig. 1). Monovalent crRNAs are referred to by their complementary length to the target DNA (i.e., crRNA-##). Bivalent crRNAs are referred to by their linker length (Linker-##).

For transcribing monovalent crRNA, two fully complementary ssDNA oligos were hybridized together in saline-sodium citrate (SSC) buffer (0.15 M NaCl, 0.015 M Na3citrate · 2H2O, pH 7.0) at a final dsDNA concentration of 25 µM by adding 5 µL of a 100 µM oligo stock, 5 µL of the 100 µM complementary oligo stock, 9 µL of H2O and 1 µL of 20xSSC buffer together in a PCR tube. DNA were hybridized by heating to 98 °C for 5 min and then cooling a degree a minute until 18 °C was reached. For transcribing bivalent crRNA, two ssDNA oligos encoding each half the crRNA were hybridized and then elongated using KAPA2G Robust PCR master mix to form fully complementary dsDNA. The thermocycler was programmed to heat to 95 °C for 3 min, and then cycling between (i) annealing at 65 °C for 15 s, (ii) extension at 72 °C for 15 s, and (iii) heating to 95 °C for 15 s for 25 cycles.

crRNA were transcribed from the dsDNA templates using the HiScribe™ T7 High yield RNA synthesis kit (NEB). Briefly, 3 µL of dsDNA (1 µg total) was added to a reaction mixture comprising T7 RNA polymerase mix (2 µL), an NTP mixture (8 µL total, 10 mM final), dithiothreitol (1 µL, 0.1 M final), and 16 µL of nanopure water, which resulted in a total reaction volume of 30 µL. The reaction was incubated at 37 °C for 4 h and then purified using the NEB Monarch RNA cleanup kit. All transcribed crRNA were characterized using a nanodrop spectrophotometer to quantify the amount of crRNA, and gel electrophoresis was used to assess length and purity. All crRNA were stored in a −20 °C freezer until needed.

### *Trans*-cleavage kinetics experiments
Cas12 *trans*-cleavage activity was determined by measuring the rate of hydrolysis of a DNA reporter substrate that was functionalized with a 5′ 6-fluorescein (56-FAM) and a 3′ Iowa Black®FQ Dark Quencher (3IABkFQ). Initially, hairpin substrate (HP) and linear substrate were compared keeping the total number of nucleotides the same (Supplementary Fig. 2). We hypothesized that decreasing the distance between fluorophore and quencher would decrease background signal, which would lead to improved sensitivity. This is an appealing strategy being pursued in the field[22,31–33]. We found that Cas12 had faster *trans*-cleavage kinetics for linear substrate and that HP was more susceptible to DNase. Therefore, linear substrate (5'-56-FAM/TTT TTA TTT TT/3IABkFQ/-3') was used for all kinetic experiments. Hydrolysis was measured by monitoring fluorescence intensity as a function of time using a BioTek Synergy H1 microplate reader equipped with a xenon light source and monochromator. Fluorescence measurements were taken every minute for 12 hours at 37 °C using excitation and emission wavelengths of 485/20 and 528/20 nm, respectively, with an instrument gain set to 80. For the 12 h measurement at 37 °C, each plate was covered. The plate reader applies heat from the top to prevent evaporation. In between measurements, plates were shaken with double orbital rotation. Data was background corrected for photobleaching by subtracting substrate only control. The resulting slope of the initial rate of reaction was used to compare crRNA activities.

The *trans*-cleavage reactions were prepared by diluting a 100 µM LbCas12a stock solution to a 1 µM stock solution using diluent supplied by the manufacturer. Then, 20 µL of the 1 µM LbCas12a stock was mixed with ~1–9 µL of crRNA depending on the crRNA stock concentration, which was then added to 10 µL of 10 × NEB r2.1 buffer diluted in ~53–57 µL of nanopure water. The ribonucleoprotein complex was formed by incubating this 88 µL mixture in a black 96-well plate for 15 min at room temperature.

Cas12-crRNA complexes were activated for *trans*-cleavage activity by adding activator dsDNA (~2 µL) to the ribonucleoprotein complexes at the desired final dsDNA concentration. The *trans*-cleavage reaction was initiated by adding 10 µL of reporter DNA resulting in a final concentration of 100 nM and a total reaction volume of 100 µL. The order of reagent addition was 10× NEB r2.1 buffer → nanopure water → LbCas12 → crRNA → activator DNA → and reporter DNA. The final reaction concentrations were 200 nM LbCas12, 12.5 nM crRNA, 100 nM reporter substrate, and the desired activator DNA concentration, respectively.

For measuring Michaelis-Menten kinetics, 50 nM Cas12, 12.5 nM crRNA, and 1 nM activator DNA was used while varying reporter substrate

concentration. The reporter substrate concentration was varied as follows: 10, 25, 50, 75, 100, 250, 500, 2000 and 4000 nM.

### LoD measurements

For sensitivity studies, a portion of the SARS-Cov-2 N-gene was cloned into a pUC19 plasmid using Gibson assembly, which is available at Addgene. The resulting plasmid was used as activator DNA. LbCas12 background activity was determined by measuring LbCas12 activity (i.e., the initial slope) in the presence of no activator DNA (Supplementary Fig. 2). The LoD was calculated by plotting a standard curve of LbCas12 activity as a function of different activator DNA concentrations for each crRNA followed by a regression analysis and calculating the standard error. The standard error was divided by the coefficient of the X-variable and multiplied by 3.3 to calculate the LoD for each crRNA. The concentrations of the activator DNA for making the calibration curve were as follows: 0.0005, 0.001, 0.005, 0.01, 0.05, 0.1, and 0.5 nM. Duplicate measurements were taken for each concentration.

### Single-nucleotide polymorphism (SNP) selectivity studies

LbCas12 selectivity was measured by comparing LbCas12 activity between a perfect match DNA sequence and sequences containing a single-point mutation. For these measurements, linear double-stranded activator DNA was used, which was amplified from IDT gBlocks encoding the perfect match target sequence and mismatch sequences containing a single-point mutation across the PAM and up to 20 nucleotides downstream, where the 20th position refers to the 3′ terminus of the target strand. PCR reactions were performed on all twenty-four gBlocks with two primers, where the reverse primer contained an overhang containing a Cas9 target DNA binding site for the bivalent crRNA studies. PCR amplicons were gel purified using the QIAquick Gel Extraction kit. Selectivity studies were carried out in duplicate for each crRNA.

### Bivalent crRNA studies

For bivalent crRNA studies, the activity of Cas12 was evaluated using the following reaction conditions: 100 nM Cas12, 100 nM dCas9, 25 nM bivalent crRNA, 1 nM activator DNA and 100 nM reporter substrate. The order of reagent addition was 10× NEB r2.1 buffer → nanopure water → LbCas12 → crRNA → dCas9 → activator DNA → and reporter DNA. For the experiments that involved both Cas12 and dCas9, LbCas12 was incubated with crRNA for about 15 min before adding dCas9 and the remaining reagents.

### Statistics and reproducibility

All statistics were carried out using Excel calculating the mean and standard deviation. All replicates were carried out as independent experiments. Error bars are included in the Figures and the number of replicates are defined in the figure captions.

### Reporting summary

Further information on research design is available in the Nature Portfolio Reporting Summary linked to this article.

## Conclusion

We quantified the kinetics of LbCas12a *trans*-cleavage kinetics for crRNAs that varied in length ranging between 15-40 nucleotide complementarity to target DNA, and studied the effects of crRNA valency. We found that crRNA-20 performed the best in terms of reaction velocity and sensitivity. Additionally, we quantified the kinetics for LbCas12a *trans*-cleavage activity for purine-pyrimidine SNP mutations for all 20 base pair target sequences of the SARS-CoV-2 N gene. We hypothesized that shorter crRNA would enable better SNP discrimination, which was the case. crRNA-15 enabled reliable discrimination of SNPs using only two measurements by comparing activities of sample to WT sequence. Furthermore, we discovered that SNP sensitivity is dependent upon the structure of the Cas12-crRNA-DNA ternary complex, where nucleotide positions most sensitive to SNP

mutations were the ones in closest contact to Cas12, which had periodicity of ~6-7 nucleotides. This corresponded to two opposite sides of the helix.

Lastly, we investigated multivalency as a strategy to enhance biosensing through cooperative binding by using long tandem crRNAs to template the assembly of bivalent Cas complexes. We showed that bivalent Cas12-dCas9 decreased background activity and had synergistic activity for a linker length of 5 ribonucleotides. Due to the profound impact CRISPR-Cas diagnostics are having for rapid point-of-care testing detection, we believe these studies will aid in designing next-generation companion diagnostics.

## Data availability

The authors confirm that all data supporting the findings of this study are available within the article and its supplementary materials. The sequence for T7 genomic DNA can be found under accession code NC_001604 at https://www.ncbi.nlm.nih.gov/nuccore/NC_001604. All raw data is compiled in the supplementary data spreadsheet.

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

## Acknowledgements
K.Y. would like to acknowledge the support from the Society for Analytical Chemists of Pittsburg and Miami University start-up funds. The authors gratefully acknowledge support from the Department of Chemistry and Biochemistry, Miami University. Schematics were made using BioRender.com.

## Author contributions
E.A., A.B. and K.Y. conceived the study; E.A. performed methodology including Michaelis-Menten and LoD experiments; A.B. performed the gene cloning; E.A. and K.Y. performed data analysis, interpretation, and wrote the manuscript; Supervision and Funding, K.Y.

## Competing interests
The authors declare no competing interests.
