## [Transparent Peer Review file · Communications Biology]

Profiling crRNA Architectures for Enhanced Cas12 Biosensing

Corresponding Author: Dr Kevin Yehl

Version 0:

Reviewer comments:

Reviewer #1

(Remarks to the Author)

In this study, Ajibode et al. investigated various aspects of crRNA design critical for Cas12a-based biosensing, including protospacer length, activator DNA concentration, and binding valency. By performing enzyme kinetics measurements, they reported that crRNA of different lengths has similar binding affinities to reporter substrates but differs in k_{cat} values. Also interestingly, they found that truncated crRNA-15 improves its specificity toward detecting SNP, which is of high clinical significance. Last, they engineered a hetero-bivalent Cas12a-Cas9 biosensor with a significantly improved high signal/background ratio. This work fits into the scope of Communications Biology, and I recommend publication after revision.

Major points:

1. The authors used crRNA-40 to optimize the equilibrium conditions for saturating DNA. In their latter study, however, it seems that crRNA-40 is not the best design despite its similar binding affinity to crRNA-20. It is unclear why they chose crRNA-40. It is also unclear why they used 1nM of activator DNA for this assay.
2. They claim a minimum of 100x of Cas12 is required to saturate target DNA. Is this statement accurate? Shouldn't this highly depend on the target DNA concentration and K_d ?
3. The authors should rationalize why they studied the crRNA longer than 25 nt. Many biochemistry or structural biology papers suggest that LbCpf1 crRNA lengths between 20 and 25 nt yield optimal cleavage efficiency. Most commercial vendors also sell crRNA in that range. I was not as surprised as the authors when they found that crRNA-20 has the fastest reaction velocity.
4. The authors ascribed the lower activity of crRNA-30 to the potential dimerization. Perhaps they can run a denaturing gel to confirm it. If indeed crRNA-30 dimerizes, the authors should consider removing the crRNA-30 data from the main figures.
5. Extended crRNA inhibits Cas9 activation. Maybe this is a potential mechanism for why crRNA-30,35,40 have lower activity. Perhaps they induced conformation changes that reduced enzyme turnover?
6. It is unclear why background fluorescence exists when there is no activator DNA. Could it be DNase contamination? What if they perform a control without crRNA?
7. The enzyme kinetics study was nicely performed. It would be even better if the authors could add crRNA-15 in comparison with crRNA-20 and crRNA-40. How will V_{max} and K_m change for truncated crRNA?
8. Replicates and error bars should be included in Figure 4(a-f). For Figure 2, how much weight do error bars hold when $n=2$?
9. The hetero-bivalent biosensor design is novel. However, proper controls should be added. It is unclear if dCas9 enhances binding through its crRNA. Even apoCas9 binds DNA non-specifically. The authors should design and use a bivalent crRNA containing a non-targeting sequence for dCas9. The result would be more convincing and support the bivalent binding model if they still observe improved SE.
10. The authors should demonstrate that their assay can detect target DNA in a mixed pool of non-target DNA(s). CRISPR proteins tend to bind nucleic acid non-specifically. Can crRNA-15 or the bivalent version of crRNA-15 demonstrate similar levels of sensitivity or selectivity? This demo will be closer to the real-world application setting.

Minor points:

1. Scheme 1b is confusing. Each enzyme protein should be labeled, and the authors should just show a bivalent design if they only tested that in the paper.
2. The LoD calculation described in the results varies slightly from what is described in the methods: multiplying by 3 vs 3.3.
3. Any explanation for the dip in fluorescence readings around 1 hour in Figure 2b?

4. The label fonts in Figure 4(a-g) are too small and hard to read. The red label in Figure 4h is not clear in combination with orange.

Reviewer #2

(Remarks to the Author)

Recommendation: This manuscript provides a valuable investigation into optimizing CRISPR-Cas12a biosensors for SNP detection by systematically analyzing the impact of crRNA architecture. The authors identify optimal crRNA lengths for different detection goals and uncover a novel position-dependent sensitivity to SNPs. They further develop a structural model to explain this phenomenon and demonstrate the synergistic activity of bivalent crRNA sensors for improved signal enhancement. These findings have significant implications for advancing point-of-care diagnostics of CRISPR-based technologies, and I recommend publishing with minor revisions.

Comments:

The assay's reporter sequence, which forms the basis of its signal, lacks important information.

- Specifically, it is unclear which black hole quencher (BHQ-1 or BHQ-2) the authors used, and details on the sequence associated with the reporter are also missing.
- Was the reporter sequence optimized for this assay or used as reported in previous studies? This is critical, as the sequence directly affects the quenching efficiency of the FAM, which in turn impacts the assay's sensitivity.
- Could the authors provide clarification on whether the distance between the quencher and fluorophore affects background levels and assay sensitivity?
- Additionally, photobleaching after 10–12 hours of measurement may alter signal stability—was this accounted for? Are there controls to correct for the rate of photobleaching, assuming it remains constant, to accurately reflect the kinetics of the reaction?

The authors report that crRNA lengths greater than 15 nucleotides of complementarity to the activator DNA were required to achieve rapid trans-cleavage activity, though it appears that 15 nucleotides still showed some level of activity. Were constructs shorter than 15 nucleotides tested to establish 15 as a benchmark, was crRNA with 12 or 10 nucleotides tested, could it be that 15 complement be an outlier rather than the definitive lower boundary?

Could the authors provide further insight/explanation into why a 5-base difference results in such a drastic change in kinetic behavior? (between 15 and 20)

The effects of linker presence and sequence complementarity are significant for certain strands length. Are there kinetic equations that could model these experimental observations? Is there a framework that could predict these interactions, potentially generalizing the findings to other DNA targets? For example, would the same linker length and design work effectively for other sequences, or would it require repeating similar studies?

Could the authors expand the discussion on the predictability of this behavior/design for various DNA targets, and provide insights on the optimal design parameters or the generalizability of this approach?

Figure 4: Very busy figure and hard to see most of the data points, maybe to enhance clarity and facilitate comparison, the data should be presented in a single, grouped bar chart. The x-axis should represent crRNA length (20, 25, 35, 40), with each length assigned a distinct color. Within each crRNA length group, individual bars should represent the different SNP mutations (PAM M1 to M20). The y-axis should clearly indicate "% Activity." This could enable straightforward comparison of the effects of crRNA length and SNP mutation on activity, highlighting trends and would help improve data interpretation. Also given that the authors have a lot of data points, histograms can be moved to SI and a heatmap might be a more effective way to visualize the trends. The x-axis could represent the SNP mutations, the y-axis the crRNA lengths, and the color intensity could represent the % Activity?

Figure 1: Consider integrating details from Figure S1 to enhance clarity and provide a comprehensive overview of the study. Emphasizing the position of the linker within the construct is also important. Focusing specifically on the description of the system while also summarizing all tested modifications can improve the figure's clarity and prevent dilution with less critical details.

Reviewer #3

(Remarks to the Author)

Version 1:

Reviewer comments:

Reviewer #1

(Remarks to the Author)

The authors have addressed our comments and thus we recommend publication. Some new results are very interesting and

exciting.

The only minor point is that LOD calculation is still incorrect (multiplied by 3).

Reviewer #2

(Remarks to the Author)

The authors have addressed many of the concerns from prior revisions, and I definitely agree that advancing CRISPR-based biosensors towards commercialization or translational research is limited by the lack of thorough investigation and empirical testing of optimal parameters. This study is valuable for designing biosensors for emerging targets by detailing what the impact of architecture and linker size and I recommend it for publication. I only have some very minor comments to address:

- Figure 2b): The legend colors do not match the plot colors, which is confusing. Please ensure they are consistent.
- Figure 2a): Consider using a ladder with better resolution or providing the original image and an enhanced-contrast version in the SI to improve visualization of the dimer mobility shift. It is currently difficult to discern band size differences due to image resolution. Improving the resolution may address this.
- Kinetics Assay: For the 12-hour measurement at 37 degrees, please clarify in the methods section how evaporation was prevented (e.g., by covering) to ensure reproducibility.
- T7 Genomic DNA Assay: The data related to the T7 genomic DNA application could be reduced to Figure 4E and potentially added to Figure 3, all the rest could be moved to SI. The authors' brief mention that T7 enhanced sensitivity (instead of reducing it) is surprising. It might reflect an impact of T7 on the FAM dye environment. Were controls performed to verify any impact on dye quantum yield (QY) in the presence or absence of T7? Given that this finding is unexpected and lacks a clear hypothesis, detailed data could be moved to the SI, as it is just there to validate existing literature.
- Figure 6: Adding a schematic to Figure 6 explaining bivalency and linker size would be beneficial (last part of SI figure S1) just like the authors did for Figure 1. Furthermore, the color coding of Figure 6-b between (+) DNA and (-) DNA plots is too similar and confusing. Using distinct symbols (e.g., triangles and circles for +/- DNA, respectively) would improve readability.
- Linker Size: While the correlation between linker size and activity is now established, the linker sequences vary in base identity, including Gs and Cs. Were these sequences designed to avoid secondary structures? I definitely understand that testing numerous linker sequence iterations is cost-prohibitive, it is not expected for this study as many parameters were already optimized, but can the authors indicate whether this was considered in the design or if it will be examined in future work.

Reviewer #3

(Remarks to the Author)

Version 2:

Reviewer comments:

Reviewer #2

(Remarks to the Author)

The authors have effectively incorporated the suggested revisions and addressed my earlier comments. The manuscript is now significantly improved, and I recommend publication.

Reviewer #1 (Remarks to the Author):

Comment 1:

The authors used crRNA-40 to optimize the equilibrium conditions for saturating DNA. In their latter study, however, it seems that crRNA-40 is not the best design despite its similar binding affinity to crRNA-20. It is unclear why they chose crRNA-40. It is also unclear why they used 1nM of activator DNA for this assay.

Response:

crRNA-40 was originally chosen because we assumed that a longer crRNA would form a weaker crRNA-Cas12 ribonucleoprotein (RNP) complex. Therefore, finding the Cas12 concentration that saturates formation of the RNP complex would also be sufficient for the shorter crRNAs.

From the reviewer's suggestion, we have tested crRNA-20 and this data is now included as part of Figure 2. This new data shows that saturation is achieved at a much lower Cas12 concentration, ~20 nM, and the K_D was measured to be 4.1 nM. This agrees well with work by Nguyen et al. (Nature Communications volume 11, Article number: 4906 (2020): Figure 2g), where they measured the K_D to be ~5 nM.

We have edited the manuscript to include this explanation, also copied below for convenience:

*crRNA-40 was selected for optimizing assay conditions because we anticipated that a longer crRNA would form a weaker complex with Cas12, thus requiring more Cas12 for saturating formation of the crRNA-complex and that this concentration would be sufficient for the shorter crRNAs. We also tested crRNA-20 to confirm this hypothesis. Results are summarized in **Figure 1b & d** and show that saturation in activity for crRNA-20 and crRNA-40 occur at ~20 nM and ~100 nM Cas12, respectively. Since activated Cas12 is dependent upon DNA binding, the data was fit to a K_D fit model (Figure 1c & e). The K_D was measured to be 4.1 nM for crRNA-20, which agrees well with work by Nguyen et al., where they measured the K_D to be ~5 nM.¹⁰ Interestingly, the fit for the crRNA-40 showed a Hill coefficient of 4.6 and a K_D of ~62 nM. We speculate that the large Hill coefficient is due to a two-step equilibrium comprising Cas12 binding both crRNA and DNA, and the K_D 's are similar.^{19,20} Based on these results, a ~100x and ~12.5x excess of Cas12 and crRNA to 1 nM activator DNA are required for saturation in Cas12 activity.*

Figure 1. Condition optimization for saturating target DNA binding by Cas12. (a) Schematic illustrating the dynamic equilibrium for forming the activated Cas12 ternary complex. (b&d) A plot summarizing Cas12 kinetics having varying Cas12 concentration and keeping crRNA (crRNA-20 & 40) and target DNA concentrations constant at 12.5 nM and 1 nM, respectively. (c&e) A plot summarizing normalized Cas12 *trans*-cleavage activity from (b&d). The data was fit to a K_d fit model (dashed grey line).

We initially chose 1 nM DNA because the K_D for the Cas12-RNP complex and target DNA has been measured to be between 0.27 – 5 nM (Nalefski et al. *Nucleic Acids Research*. 52, 4502-4522, 2024). Using 12.5 nM crRNA and assuming crRNA-Cas12 RNP complex formation is saturated based on Figure 2 data, this equates to 71 - 98% bound target DNA. Since Cas12 *trans*-cleavage activity is linear in response to DNA concentration (Fig. 2d), we believe DNA binding is saturated.

Higher concentrations of crRNA-Cas12 and DNA can be tested, which would increase the dynamic range of the assay, but this also comes as a tradeoff for cost. In addition, we believe that higher DNA concentrations are not representative of real-world applications. We edited the manuscript to be more accurate including this discussion. We have edited the manuscript to include this explanation, also copied below for convenience:

Target DNA was set to 1 nM because we assumed this would be the upper limit for detection for a typical assay and that higher DNA concentrations would not be representative of real-world samples. Additionally, the K_D for the Cas12-crRNA ribonucleoprotein complex binding target DNA has been measured to be between 0.27 – 5.0 nM¹⁸, so conditions that saturate crRNA-Cas12 complex formation using 12.5 nM crRNA would equate to 71 – 98% bound target DNA. Higher concentrations of crRNA-Cas12 and DNA would increase the dynamic range of the assay, but this will come as a tradeoff for cost.

We originally thought that under these conditions, saturation in Cas12 activity as a function of Cas12 concentration would imply saturation in DNA binding. However, upon preparing our response, we realized that this only implies crRNA is saturated by Cas12 binding. We have revised the text to be more accurate.

Comment 2:

They claim a minimum of 100x of Cas12 is required to saturate target DNA. Is this statement accurate? Shouldn't this highly depend on the target DNA concentration and K_D ?

Response:

We thank the reviewer for bringing this to our attention. Our original statement was not accurate. We agree with the reviewer that saturation of DNA binding is dependent on K_D and the concentration of Cas12-crRNA complex, when keeping DNA concentration constant. Based on previously measured binding affinity between crRNA-Cas12 and target DNA, and because we observe a linear response as a function of DNA concentration, we believe DNA binding is saturated or nearly saturated.

We have amended the text to be more accurate to say:

Based on these results, a minimum of ~100x and ~12.5x excess of Cas12 and crRNA to 1 nM activator DNA, respectively, are required for saturation in Cas12 activity.

Comment 3:

The authors should rationalize why they studied the crRNA longer than 25 nt. Many biochemistry or structural biology papers suggest that LbCpf1 crRNA lengths between 20 and 25 nt yield optimal cleavage efficiency. Most commercial vendors also sell crRNA in that range. I was not as surprised as the authors when they found that crRNA-20 has the fastest reaction velocity.

We agree that many biochemistry or structural biology papers suggest that LbCpf1 crRNA lengths between 20 and 25 nt yield optimal 'cis' cleavage (i.e., cleavage of the target DNA). However, we aim to quantify the optimal length for trans-cleavage. The only other report that we are aware of that studied dependency of crRNA length on trans-cleavage is work by Nguyen et al. (Nature Communications volume 11, Article number: 4906 (2020): Figure 1d). However, they showed that longer crRNA lengths up to a spacer length of 39 (i.e., using our nomenclature, crRNA-39) increased activity ~3.5x compared to a crRNA spacer length of 20 (i.e., crRNA-20). We believe this difference is interesting, and worth investigating further, but is beyond the scope of this study. We also wanted to know the optimal crRNA length for trans-cleavage activation because this would guide bivalent studies.

We have edited the text to include this comparison, also copied below for convenience:

Surprisingly, crRNA-20 resulted in the fastest reaction velocity, where longer crRNAs decreased in activity. This differs from what Nguyen et al. observed, where longer crRNAs resulted in 3.5x increased activity compared to crRNA with a spacer length of 20 (i.e., crRNA-20).

Comment 4:

The authors ascribed the lower activity of crRNA-30 to the potential dimerization. Perhaps they can run a denaturing gel to confirm it. If indeed crRNA-30 dimerizes, the authors should consider removing the crRNA-30 data from the main figures.

We thank the reviewer for this suggestion. We did run a denaturing gel and observed the ‘dimerized’ band go away. Please see the gel image ‘b’ below. We have included this new data in the text as Supplementary Figure S3.

Figure S3. Denaturation gel to confirm dimerization of crRNA-30. (a) Gel image before denaturation (b) Gel image of denatured gel. Lane 1 represents the 50 bp DNA ladder, lanes 2 – 6 represent the different length of monovalent crRNAs ranging from crRNA-15 to 40 in the order 38, 43, 48, 53, and 58 ribonucleotides respectively in (a) and (b). The gel shows a slight change in band sizes across the different crRNAs.

We would like to include crRNA-30 data because we believe this highlights the nuances of working with nucleic acids, that sequence and length have unexpected impact on structure, and that it is important to always run full characterization (non-denaturing and denaturing) of samples. Even though we included the data, we omitted crRNA-30 when observing general trends in activity for discussions.

Comment 5:

Extended crRNA inhibits Cas9 activation. Maybe this is a potential mechanism for why crRNA-30,35,40 have lower activity. Perhaps they induced conformation changes that reduced enzyme turnover?

We agree with the reviewer that the extended crRNA reduces Cas12 trans-cleavage turnover and that this is the explanation for reduced activity. This was the conclusion from our Michaelis-Menten kinetic studies (lower V_{max} compared to similar K_m). We have modified the

text to clarify this explanation and to also mention that the reduced turnover could be due to induced conformational change in Cas12, similar to how Cas9 is inactivated by extended crRNA. However, this confirmation change is different from the induced confirmational change required for trans-cleavage activation. We thank the reviewer for this suggestion, which we believe strengthens our discussion.

This reduced turnover could potentially be due to induced conformational change in Cas12, similar to how Cas9 is sometimes inactivated by extended crRNA.

Comment 6:

It is unclear why background fluorescence exists when there is no activator DNA. Could it be DNase contamination? What if they perform a control without crRNA?

We weren't sure if the reviewers were talking about monovalent crRNA or bivalent crRNA, so we ran controls for both samples. We tested for DNase activity by measuring the cleavage of the reporter DNA substrate in the absence of target DNA by comparing Cas12 only (without crRNA) vs. crRNA only vs. crRNA+Cas12 for crRNAs: crRNA-20, Linker-0, and Linker-15. The results show that we do not have DNase contamination (i.e., activity for crRNA only or Cas12 only), though we do have high background activity for the bivalent crRNA complex.

Figure S9. Investigating background activity with no DNA. (a) Cas12 activity with and without crRNA shows no background activity in the absence of target DNA. (b) Cas12 activity with and without bivalent crRNA, linker-0. The purple curve shows a high background activity without DNA which is the same as what we see in (c) with linker-15.

Thinking about this result more deeply, we were concerned that contamination from DNA template from the transcription reaction was a possible cause for high background activity (i.e., target DNA without a PAM), as this crRNA was made through a different method as the monovalent crRNA. To test this, we added template DNA with crRNA-20 + Cas12 at varying concentrations. We observed see very little trans-cleavage activity. This data is now included as Figure S10. We also amended the text to include this discussion. Both copied below for convenience.

Figure S10. Investigating background activity with different DNA. Plots showing Cas12 trans-cleavage kinetics for (a) fully complementary target DNA, (b) hybridized DNA used for making bivalent crRNA template for transcription, and (c) fully extended DNA used for bivalent crRNA transcription (orange is PAM, purple is Cas12 crRNA, red is linker, and blue is Cas9 crRNA). Though activity is observed for the extended DNA, it is not as much as is observed in the bivalent crRNA only (Figure S8).

However, background activity was significantly higher compared to monovalent crRNA (Figure 5b, lighter colored curves). To confirm that the high background activity wasn't DNase contamination or template contamination from the in vitro transcription reaction, we tested cleavage activity of the reporter DNA substrate in the absence of target DNA by comparing Cas12 only (without crRNA) vs. crRNA only vs. crRNA+Cas12 for crRNAs: crRNA-20, Linker-0, and Linker-15. We also tested activation of crRNA-20 + Cas12 with varying concentrations of template DNA used for the in vitro transcription reaction (Figure S10). Together, the results show that we do not have DNase contamination nor does template DNA activate Cas12 to the same level observed, though we do have high background activity for the bivalent crRNAs.

Comment 7:

The enzyme kinetics study was nicely performed. It would be even better if the authors could add crRNA-15 in comparison with crRNA-20 and crRNA-40. How will V_{max} and K_m change for truncated crRNA?

We thank the reviewer for this suggestion and have repeated the enzyme kinetics studies including crRNA-15 in this set. The text includes the updated data, which is also shown on the next page for convenience. We found that both K_m and k_{cat} are significantly compromised for crRNA-15, where for crRNA-40, the main contribution for decreased activity is from reduced Cas12 trans-cleavage turnover (k_{cat}), though K_m isn't insignificant. For crRNA-15, we believe DNA binding is saturated, indicated by linear response to varying DNA concentration (Figure 2d), but is unable to induce a conformational change required for activating trans-cleavage activity. We believe crRNA-40 has slight reduced binding to reporter DNA by ~30%, which can be attributed to increased anionic repulsion between the complex and reporter substrate. crRNA-40 also has significantly reduced turnover, most likely from a different induced conformational change required for trans-cleavage.

The results show that for crRNA-15, both K_m and V_{max} are significantly compromised. We believe that crRNA-15 does saturate DNA binding, indicated by linear response to varying DNA concentration (Figure 2d), but is unable to induce a conformational change required for activating significant trans-cleavage activity. Whereas, for crRNA-40, the main contribution for

decreased activity is from reduced Cas12 trans-cleavage turnover (k_{cat}), though K_m isn't insignificant. We believe crRNA-40 has slight reduced binding to reporter DNA by ~30%, determined by comparing K_m 's between crRNA-20 and crRNA-40. We believe this is due to increased anionic repulsion. However, the max velocity varied significantly between the two crRNAs, differing by approximately 4-fold, where crRNA-20 had the fastest V_{max} . Together, these results show that Cas12 complexed with different length crRNAs have varying binding affinities for reporter substrate, but significantly differ in enzymatic turnover (k_{cat}).

continued from response to Comment 7.

Figure 3. Michaelis-Menten kinetics for Cas12 *trans*-cleavage activity for crRNA-15, 20 and -40. A summary of Cas12 *trans*-cleavage kinetics for crRNA-15 (a) crRNA-20 (b) and crRNA-40 (c) with varying substrate concentration. The concentrations of Cas12, crRNA and DNA were 50 nM, 12.5 nM, and 1 nM respectively. (d, e & f) Michaelis-Menten plots generated from (a), (b) and (c), respectively. The inset shows the fitted Michaelis-Menten parameters resulting in the best fit ($K_m = 2224$ nM, 360 nM and 536 nM for crRNA-15, crRNA-20 and crRNA-40 respectively).

Comment 8:

Replicates and error bars should be included in Figure 4(a-f). For Figure 2, how much weight do error bars hold when $n=2$?

We thank the reviewer for this suggestion. This is an extremely laborious experiment, requiring a total of 150 reactions per n . Since the general trend in activity is similar for each crRNA, we believe that this is similar to a replicate (i.e., ' n '=6), which is why we compiled the data for Figure 4g (now Figure 5g in the amended text). Due to limitation in resources, we hope the reviewer agrees that this is sufficient.

Regarding Figure 2, *n* of 2: We have compiled our data for crRNA-20 from two independent experiments, from two different batches of Cas12 shown below (*n* = 4).

Figure S4. Comparison of Cas12 and crRNA activity from different experiments. Data showing Cas12 activity from using the different batches of Cas12 and crRNA from two independent experiments. Standard deviation was calculated using $n=2$

Standard deviation and confidence intervals work for small sample numbers ($n = 2$), but standard deviation is slightly underestimated. However, this discrepancy is small compared to random variability inherent to collecting tiny data sets. Since the general trend holds for each crRNA at each DNA concentration (i.e., crRNA-20 > crRNA-25 > crRNA-35...), along with our compiled data above, we have confidence in our result. We have amended the text to emphasize that the standard deviation is calculated from an $n = 2$, and that this slightly underestimates standard deviation. The amended text is copied below for convenience.

It is important to note that the standard deviation is calculated from an $n = 2$, which slightly underestimates standard deviation, but the general trend of crRNA-20 > crRNA-25 > crRNA-35 > crRNA-40 > crRNA-30 > crRNA-15 holds true for each DNA concentration (Figure dd). We also compiled data from crRNA-20 from two batches of Cas12 ($n = 4$), shown in Figure S4, which shows high reproducibility.

Comment 9:

The hetero-bivalent biosensor design is novel. However, proper controls should be added. It is unclear if dCas9 enhances binding through its crRNA. Even apoCas9 binds DNA non-specifically. The authors should design and use a bivalent crRNA containing a non-targeting sequence for dCas9. The result would be more convincing and support the bivalent binding model if they still observe improved SE.

Based on the reviewer's suggestion, we repeated the bivalent crRNA studies, but included a 'monovalent' DNA template and compared activity to 'bivalent' DNA template. We also compared activity with and without Cas9. The monovalent DNA is the same overall length as the bivalent DNA template and serves as the suggested control of a bivalent crRNA containing a non-targeting

sequence for dCas9. This simplifies having to make an entire set of bivalent crRNAs, and only requires testing an additional target DNA.

The new data are shown below for each bivalent crRNA varying in linker length, where we measured signal enhancement (SE) by taking the difference in fluorescence between (i) bivalent DNA vs monovalent DNA having both Cas12 and Cas9; (ii) bivalent DNA with Cas12 and Cas9 vs without Cas9; and (iii) bivalent DNA vs monovalent DNA having Cas12 and no Cas9.

We found really interesting results that confirms synergistic activity for a Linker-5. Only in the presence of a bivalent target DNA, Cas12, and Cas9 is enhanced activity observed (**Figure 6d & 6e**). Comparing bivalent versus monovalent DNA without Cas9 shows no enhancement (**Figure 6f**). This shows that bivalency enhances activity, but is distance dependence. The distance requirement is approximately 5 RNA nucleotides in the linker sequence to span 20 base pair DNA, as the latter is the distance between the Cas12 and Cas9 binding sites.

We have updated Figure 6 to include this data in the amended text, copied below for convenience.

Figure 6. Cas12 activity with bivalent crRNA. (d) Cas12 activation in the presence of dCas9 with bivalent and monovalent DNA (e) Cas12 activation in the presence and absence of dCas9 with bivalent DNAs only (f) Cas12 activation in the absence of dCas9 with monovalent and bivalent DNA.

*we continued to explore how the addition of dCas9 altered binding and subsequent Cas12 activation to activator DNA containing two corresponding binding sites. The data are summarized in **Figure 6** for each bivalent crRNA varying in linker length, where we compared fluorescence by taking the difference between (i) bivalent DNA vs monovalent DNA having both Cas12 and Cas9; (ii) bivalent DNA with Cas12 and Cas9 vs without Cas9; and (iii) bivalent DNA vs monovalent DNA with Cas12 and no Cas9. Remarkably, the addition of dCas9 drastically reduced background activity, while only slightly decreasing overall activity in the presence of activator DNA. This resulted in a signal enhancement improvement of ~360% when comparing Linker-20 in the presence and absence of dCas9, where only ~144% signal enhancement was observed for monovalent crRNA-20 (**Figure 6c**).*

*We also found really interesting results that show synergistic activity for a Linker-5. Only in the presence of a bivalent target DNA, Cas12, and Cas9 is enhanced activity observed (**Figure 6d & e**). Comparing bivalent versus monovalent DNA without Cas9 shows no enhancement (**Figure 6f**). This data shows that bivalency enhances activity, but is distance dependence. The distance requirement is approximately 5 RNA nucleotides in the linker to span 20 base pair extraneous DNA, as the latter is the distance between the Cas12 and Cas9 binding sites.*

Comment 10:

The authors should demonstrate that their assay can detect target DNA in a mixed pool of non-target DNA(s). CRISPR proteins tend to bind nucleic acid non-specifically. Can crRNA-15 or the bivalent version of crRNA-15 demonstrate similar levels of sensitivity or selectivity? This demo will be closer to the real-world application setting.

This is a very interesting suggestion. We carried out the suggested experiment using T7 phage genomic DNA as the ‘mixed pool of non-target DNA(s)’. We tested 0x, 1x, 5x, and 10x excess by mass and found no difference in activity. In fact, there is a general trend for slight increase in sensitivity when T7 genomic DNA is added. This data shows that Cas12 is very selective for its target, and is also very selective for ssDNA for trans-cleavage. This corroborates previous work by Nalefski et al. (iScience. 2021 Aug 18;24(9):102996. doi:) and Chen et al. (Biochemistry. 2020 Apr 7;59(15):1474–1481) which show Cas12 preferentially cleaves ssDNA over dsDNA for trans-cleavage.

Figure 4. Performance of Cas12-crRNA-20 and target DNA in a mixed pool of non-target DNA. Cas12 activity (a) with target DNA only, (b) with 1x T7 genomic DNA, (c) 5x T7 genomic DNA, and (d) 10x T7 genomic DNA. (e) Summary of sensitivity studies showing no significant change in Cas12 activity. Error bars show standard deviation (n=2).

Next, to simulate detection similar to a real-world setting, we tested the ability of crRNA-20 to target DNA in a mixed pool of non-target DNA using T7 genomic DNA as the mixed pool DNA and varying up to 10x excess by mass. Results are summarized in **Figure 4**, where sensitivity is unaffected. Surprisingly, we observed a general trend for increased sensitivity. This corroborates seminal work by Chen et al. and work by Nalefski and Smith et al., which shows Cas12 preferentially targets ssDNA for trans-cleavage over dsDNA.^{5,21,22}

Minor points

1. Scheme 1b is confusing. Each enzyme protein should be labeled, and the authors should just show a bivalent design if they only tested that in the paper.

We have corrected Scheme 1b to more accurate. Specifically, we only show the monovalent and bivalent design, including the hetero-bifunctional Cas12-Cas9 crRNA for the bivalent design.

2. The LoD calculation described in the results varies slightly from what is described in the methods: multiplying by 3 vs 3.3.

We thank the reviewer for catching this typo. We indeed multiplied 3.3 and have corrected the text to reflect this.

3. Any explanation for the dip in fluorescence readings around 1 hour in Figure 2b?

We believe the dip in fluorescence is due to instrumental error caused by vibrations from our centrifuge. The plate reader is on the same bench as the centrifuge. We have added new data that does not have this noise, which is included in the updated Figure 2.

4. The label fonts in Figure 4(a-g) are too small and hard to read. The red label in Figure 4h is not clear in combination with orange

We have increased the font size and made the figure fit the entire width of the page to improve readability. We also changed the coloring in Figure 4h to improve visibility of the mutations. We changed the crRNA to be gray instead of orange, and the mutations to be red.

Reviewer #2 (Remarks to the Author):

Comment 1:

The assay's reporter sequence, which forms the basis of its signal, lacks important information.

Sorry, this was a mistake. We thank the reviewer for bringing this to our attention. The sequence is now added to methods and to the supplemental in Table S1, also copied below for convenience.

5'- /56-FAM/TTT TTA TTT TT/3IABkFQ/ -3'

Comment 2:

Specifically, it is unclear which black hole quencher (BHQ-1 or BHQ-2) the authors used, and details on the sequence associated with the reporter are also missing.

Sorry, we had a typo in the text. We used Iowa Black®FQ (IABkFQ) as the quencher. IABkFQ is similar to BHQ-1 and has broad absorbance ranging from 420 to 620 nm, which is ideal for quenching FAM6. This information is now included in Table S1.

Comment 3:

Was the reporter sequence optimized for this assay or used as reported in previous studies? This is critical, as the sequence directly affects the quenching efficiency of the FAM, which in turn impacts the assay's sensitivity.

We originally tried to optimize the sequence to use a hairpin (HP) reporter to improve quenching efficiency, but discovered that Cas12 had faster trans-cleavage kinetics for linear substrate and that HP has higher sensitivity to DNase. This data is shown below (Figure S2a and b, respectively). Therefore, we thought it was best to continue with the ssDNA polyT design. We have included this data in the amended text as Supplemental Figure 2, and added a brief discussion.

Figure S2: Trans-cleavage activity of DNA reporter substrates. (a) Cas12 trans-cleavage activity between linear substrate and (b) hairpin substrate showed that Cas12 cleaved the linear substrate better than the hairpin substrate. (c) Shows the effect of DNase on both DNA reporter substrates with hairpin substrate having a lower background fluorescence yet increased signal in comparison to the linear substrate.

Cas12 trans-cleavage activity was determined by measuring the rate of hydrolysis of a DNA reporter substrate that was functionalized with a 5' 6-fluorescein (56-FAM) and a 3' Iowa Black®FQ Dark Quencher (IABkFQ). Initially, hairpin substrate (HP) and linear substrate were compared keeping the total number of nucleotides the same (Figure S2). We hypothesized that decreasing the distance between fluorophore and quencher would decrease background signal, which would lead to improved sensitivity. This is an appealing strategy being pursued in the field.¹⁵⁻¹⁸ We found that Cas12 had faster trans-cleavage kinetics for linear substrate and that HP was more susceptible to DNase. Therefore, linear substrate (5'-56-FAM/TTT TTA TTT TT/3IABkFQ/-3') was used for all kinetic experiments.

Comment 4:

Could the authors provide clarification on whether the distance between the quencher and fluorophore affects background levels and assay sensitivity?

Yes, the distance between the quencher and the fluorophore affects background levels. A closer distance results in better quenching efficiency, which has an r^6 dependency, where r is distance. Decreasing distance between fluorophore and quencher will decrease background,

which would also lead to a better limit-of-detection. We originally tried optimizing the reporter substrate, but discovered that the hairpin reporter was more DNase sensitive, so didn't pursue further. In our current design, our quenching efficiency was measured to be ~96%, so we can potentially improve sensitivity further through using different reporter design. Multiple groups have pursued this strategy: Smith et al. (Biochemistry 2020, 59, 1474–1481) where they engineered a nicked substrate; Rossetti et al. where they engineered a hairpin substrate to enhance Cas12a trans-cleavage activity (Nucleic Acids Research, 2022, Vol. 50, No. 14 8377–8391); work by Fu et al. where they functionalized gold nanoparticles with reporter substrate to improve quenching efficiency further (Anal. Chem. 2021, 93, 4967-4974); and work by Lee et al. where they engineered a highly efficient DNA reporter, TATT-5C (Biochip J. 2022 Sep 13;16(4):463–470. doi: 10.1007/s13206-022-00081-0).

However, the goal of this work isn't to be the most sensitive Cas12 sensor, rather (i) to better understand how crRNA affects Cas12 activation and trans-cleavage activity and (ii) to test for synergistic activity for bivalent crRNA.

We do agree with the reviewer that this discussion on optimizing assay sensitivity based on reporter design is important, so we amended the text to reflect this and cited the work above.

Initially, hairpin substrate (HP) and linear substrate were compared keeping the total number of nucleotides the same (Figure S2). We hypothesized that decreasing the distance between fluorophore and quencher would decrease background signal, which would lead to improved sensitivity. This is an ongoing and appealing strategy being pursued in the field.¹⁵⁻¹⁸

Comment 5:

Additionally, photobleaching after 10–12 hours of measurement may alter signal stability—was this accounted for? Are there controls to correct for the rate of photobleaching, assuming it remains constant, to accurately reflect the kinetics of the reaction?

All samples were background subtracted with substrate to account for photobleaching. We have amended the text to include this detail in the methods, also copied below for convenience.

Data was background corrected for photobleaching by subtracting substrate only control. The resulting slope of the initial rate of reaction was used to compare crRNA activities.

Comment 6:

The authors report that crRNA lengths greater than 15 nucleotides of complementarity to the activator DNA were required to achieve rapid trans-cleavage activity, though it appears that 15 nucleotides still showed some level of activity. Were constructs shorter than 15 nucleotides tested to establish 15 as a benchmark, was crRNA with 12 or 10 nucleotides tested, could it be that 15 complement be an outlier rather than the definitive lower boundary?

We have tested other target DNA sequences having decreasing complementary to target DNA for crRNA-20. We found that activity begins to drop off after 18 nucleotides in complementary

and reaches background levels of activity for 14. 15 shows detectable, but very little activity. Importantly, 15 fits within this trend so is not an outlier.

Comment 7:

Could the authors provide further insight/explanation into why a 5-base difference results in such a drastic change in kinetic behavior? (between 15 and 20)

We believe that the main reason for the difference in activity is that crRNA-15 is unable to induce a conformational change required to induce trans-cleavage activity. Another explanation could be due to weakened binding and not saturating DNA binding. However, the linear response as a function of DNA concentration rules against the latter explanation. We have amended the text to include this discussion.

Comment 8:

The effects of linker presence and sequence complementarity are significant for certain strands length. Are there kinetic equations that could model these experimental observations? Is there a framework that could predict these interactions, potentially generalizing the findings to other DNA targets? For example, would the same linker length and design work effectively for other sequences, or would it require repeating similar studies?

This is a very interesting point. In our current design, we tried to design a linker with little to no sequence complementarity to target DNA and have minimal secondary structure. We believe that this is important when designing bivalent crRNA. We also believe that optimal linker length will be dependent upon distance between the Cas12 and Cas9 binding sites. Our new results characterizing the bivalent crRNA with recommended controls from Reviewer 1 suggest that 5 ribonucleotides in the linker is optimal for spanning a 20 base pair gap between CRISPR-Cas binding sites. This data is now included in updated Figure 6. We have also included this discussion in the amended text. Both copied below for convenience. We originally designed a linker using polyU. However, since Cas12 is known to have binding interaction with polyT, we wanted to minimize any unintended interactions.

The new data are shown below, where we compared fluorescence by taking the difference between various experimental conditions (bivalent DNA vs monovalent DNA; with and without

Cas9) for bivalent crRNA having varying linker lengths. We found really interesting results that confirms synergistic activity for Linker 5. Only in the presence of a bivalent target DNA, Cas12, and Cas9 is enhanced activity observed. Comparing bivalent versus monovalent DNA without Cas9 shows no enhancement (Figure 6e). This shows that bivalency enhances activity, but is distance dependence. The distance requirement is approximately 5 nucleotide RNA linker to a 20 base pair DNA duplex, as the latter is the distance between Cas12 and Cas9 binding sites.

Figure 6. Cas12 activity with bivalent crRNA. (d) Cas12 activation in the presence of dCas9 with bivalent and monovalent DNA (e) Cas12 activation in the presence and absence of dCas9 with bivalent DNAs only (f) Cas12 activation in the absence of dCas9 with monovalent and bivalent DNA.

we continued to explore how the addition of dCas9 altered binding and subsequent Cas12 activation to activator DNA containing two corresponding binding sites. The data are summarized in Figure 6 for each bivalent crRNA varying in linker length, where we compared fluorescence by taking the difference between (i) bivalent DNA vs monovalent DNA having both Cas12 and Cas9; (ii) bivalent DNA with Cas12 and Cas9 vs without Cas9; and (iii) bivalent DNA vs monovalent DNA with Cas12 and no Cas9. Remarkably, the addition of dCas9 drastically reduced background activity, while only slightly decreasing overall activity in the presence of activator DNA. This resulted in a signal enhancement improvement of ~360% when comparing Linker-20 in the presence and absence of dCas9, where only ~144% signal enhancement was observed for monovalent crRNA-20 (Figure 6c).

We also found really interesting results that show synergistic activity for a Linker-5. Only in the presence of a bivalent target DNA, Cas12, and Cas9 is enhanced activity observed (Figure 6d & e). Comparing bivalent versus monovalent DNA without Cas9 shows no enhancement (Figure 6f). This data shows that bivalency enhances activity, but is distance dependence. The distance requirement is approximately 5 RNA nucleotides in the linker to span 20 base pair extraneous DNA, as the latter is the distance between the Cas12 and Cas9 binding sites.

Comment 9:

Could the authors expand the discussion on the predictability of this behavior/design for various DNA targets, and provide insights on the optimal design parameters or the generalizability of this approach?

This is a great suggestion. We have amended the discussion to include, also copied below for convenience.

It will be important to determine the generalizability of this approach. We believe that shorter crRNAs will always be ideal for detecting SNPs, but determining the optimal length currently

requires empirical testing. A generalizable model linking thermodynamic parameters and induced conformational changes in Cas12 with Cas12 trans-cleavage kinetics will be valuable for readily designing biosensors for emerging threats. Such a model can help explain differences observed for longer crRNAs, where we observed decreased activity and Nguyen et al. observed enhanced activity.¹⁰ Furthermore, it will be important to systematically investigate the distance dependence for multivalent CRISPR biosensors, as distance between CRISPR-Cas binding sites within a gene target is specific to the gene. Though, great efforts have been made to engineer Cas9 variants to recognize an array of PAMs.²⁹ Lastly, bivalent CRISPR-Cas biosensors offer unique programmability because the binding module (dCas9) is decoupled from the biosensing module (Cas12). This offers a novel approach to biosensor design, with hopes of unlocking cooperativity to rationally improve biosensor function.³⁰

Comment 10:

Figure 4: Very busy figure and hard to see most of the data points, maybe to enhance clarity and facilitate comparison, the data should be presented in a single, grouped bar chart. The x-axis should represent crRNA length (20, 25, 35, 40), with each length assigned a distinct color. Within each crRNA length group, individual bars should represent the different SNP mutations (PAM M1 to M20). The y-axis should clearly indicate "% Activity." This could enable straightforward comparison of the effects of crRNA length and SNP mutation on activity, highlighting trends and would help improve data interpretation. Also given that the authors have a lot of data points, histograms can be moved to SI and a heatmap might be a more effective way to visualize the trends. The x-axis could represent the SNP mutations, the y-axis the crRNA lengths, and the color intensity could represent the % Activity?

We thank the reviewer for this suggestion. We have tried a few different iterations of reformatting this data, please see below. In the first iteration, it seems that the grouped data is too busy to compare. We tried overlaying the histograms to ‘declutter’ the figure and improve interpretation, but due to six different crRNA’s being tested, some of the data is hidden, or not clearly shown in our opinion.

We did consider showing the data as a heat map, but believe that important aspects of the data are lost in the reduced dimensionality. For example, there is a sinusoidal-like trend in the data that isn’t apparent in a heat map form. Please see below on the next page. We have included the heat map in the supplemental.

To improve readability of the figure, we have increased the font and made the figure larger.

Comment 11:

Figure 1: Consider integrating details from Figure S1 to enhance clarity and provide a comprehensive overview of the study. Emphasizing the position of the linker within the construct is also important. Focusing specifically on the description of the system while also summarizing all tested modifications can improve the figure’s clarity and prevent dilution with less critical details.

We thank the reviewer for this suggestion and have modified Scheme 1 to include the varying linker lengths. We also modified the figure to not show the ternary CRISPR-Cas variant since that wasn’t tested in this work.

Reviewer #1 (Remarks to the Author):

The authors have addressed our comments and thus we recommend publication. Some new results are very interesting and exciting.

We thank the reviewer for this positive response.

Comment 1: The only minor point is that LOD calculation is still incorrect (multiplied by 3).

We apologize for the confusion, and we thank the reviewer for catching this typo. We used 3.3 for our calculations and also described this in the Results. However, we did not correct this typo in the Methods. This is now corrected. We have updated the manuscript as follows:

“The standard error was divided by the coefficient of the X-variable and multiplied by 3.3 to calculate the LoD for each crRNA.”

Reviewer #2 (Remarks to the Author):

The authors have addressed many of the concerns from prior revisions, and I definitely agree that advancing CRISPR-based biosensors towards commercialization or translational research is limited by the lack of thorough investigation and empirical testing of optimal parameters. This study is valuable for designing biosensors for emerging targets by detailing what the impact of architecture and linker size and I recommend it for publication. I only have some very minor comments to address:

We thank the reviewer for this positive response and agree such studies will help advanced CRISPR-based biosensors towards commercialization and translational research.

Comment 1:

Figure 2b): The legend colors do not match the plot colors, which is confusing. Please ensure they are consistent.

We thank the reviewer for catching this formatting error. In Figure 2b (now Figure 3b), we have updated the legend colors to match the plot colors.

Comment 2:

Figure 2a): Consider using a ladder with better resolution or providing the original image and an enhanced-contrast version in the SI to improve visualization of the dimer mobility shift. It is currently difficult to discern band size differences due to image resolution. Improving the resolution may address this.

We have provided the original gel images in the SI. We observe a larger size for crRNA-30 relative to the other crRNA in non-denaturing conditions, which is abolished when performing gel electrophoresis under denaturing conditions. We hope that these images are able to address this concern.

Comment 3:

- Kinetics Assay: For the 12-hour measurement at 37 degrees, please clarify in the methods section how evaporation was prevented (e.g., by covering) to ensure reproducibility.

We thank the reviewer for this additional comment. We have updated the text to reflect that we covered the 96-well plates and that heating occurred from both the bottom and top of the plate to prevent evaporation.

“For the 12 hours measurement at 37 °C, each plate was covered and the plate reader applies heat form the top to prevent evaporation.”

Comment 4:

- T7 Genomic DNA Assay: The data related to the T7 genomic DNA application could be reduced to Figure 4E and potentially added to Figure 3, all the rest could be moved to SI. The authors' brief mention that T7 enhanced sensitivity (instead of reducing it) is surprising. It might reflect an impact of T7 on the FAM dye environment. Were controls performed to verify any impact on dye quantum yield (QY) in the presence or absence of T7? Given that this finding is unexpected and lacks a clear hypothesis, detailed data could be moved to the SI, as it is just there to validate existing literature.

We thank the reviewer for this comment. We moved this data to Supplemental. We agree that the detailed data does not belong in the text and Figure 4e isn't enough as a standalone figure, though the experiment is very interesting and we thank the reviewer for the suggestion.

Though we think it is interesting and a little surprising, the observed increase in activity at higher T7 genomic DNA is within error, so we did not make any concrete conclusions. Importantly, we observed no background activity in the absence of target DNA. This highlights the selectivity of Cas12. Since all samples have the same starting and end-point fluorescence, ~6,000 RFUs for end-point, T7 DNA is not having any impact on dye quantum yield. We also have other data from another project measuring trans-cleavage kinetics where large amounts of target dsDNA relative to the reporter substrate does not decrease Cas12 activity. We believe this Cas12 variant has a strong preference for ssDNA for trans-cleavage.

Comment 5:

• Figure 6: Adding a schematic to Figure 6 explaining bivalency and linker size would be beneficial (last part of SI figure S1) just like the authors did for Figure 1. Furthermore, the color coding of Figure 6-b between (+) DNA and (-) DNA plots is too similar and confusing. Using distinct symbols (e.g., triangles and circles for +/- DNA, respectively) would improve readability.

We thank the reviewer for pointing this out. We have now included a schematic to Figure 6 to clarify the figure, similar to what we have in Figure S1. For Figure 6b, we have also improved the color coding between (+) DNA and (-) DNA and changed the symbols to better differentiate the two.

Comment 6:

- **Linker Size:** While the correlation between linker size and activity is now established, the linker sequences vary in base identity, including Gs and Cs. Were these sequences designed to avoid secondary structures? I definitely understand that testing numerous linker sequence iterations is cost-prohibitive, it is not expected for this study as many parameters were already optimized, but can the authors indicate whether this was considered in the design or if it will be examined in future work.

We originally were going to use a polyU linker, but were concerned that Cas12 would have a preference to polyU since the PAM is a short polyT. This concern may not have been warranted. Therefore, when designing the linker sequences, we tried to minimize any interaction with the target DNA by inputting a random sequence and comparing homology.

We did not look into secondary structure, which is a very good point that the reviewer raises. Therefore, we re-analyzed our sequences looking at potential secondary structures for the entire crRNA and for the linker region only. This is now included in the supplementary as Supplementary Figure 13 and copied below for convenience. Interestingly, bivalent crRNA L05-L20 possess very similar, nearly identical global structures, where the region next to the

longest stem loop has increasing length for increasing linker length, indicating increased distance between Cas9 and Cas12. Additionally, only L20 showed secondary structure in the linker region only, where all others do not show secondary structure in the linker. However, we believe we should not read too deeply into these results/structures, since it is not known how binding to Cas9 and Cas12 alter the secondary structure. Since we observed two separate conditions that have enhanced activity: Comparing monovalent DNA to bivalent DNA, and with and without Cas9 using bivalent target DNA, we believe this data highly suggests distance dependence synergy. We agree with the author that delving into investigating how secondary structure potentially alters synergy is very important direction of study, but believe that is beyond the scope of this work and is an ongoing active project in the lab. We thank the reviewer again for their positive feedback and suggestions, which have strengthened the paper.

Supplementary Figure 13. Predicted bivalent crRNA secondary structures. (a) Secondary structure of bivalent crRNA with no linker or L0. (b) Secondary structure of bivalent crRNA with L5. (c) Secondary structure of bivalent crRNA with L10. (d) Secondary structure of bivalent crRNA with L15. (e) Secondary structure of bivalent crRNA with L20. RNA structures were generated through RNAfold webserver.

Elizabeth Ajibode et al. Response to Reviewers

Reviewer #3

Co-reviewer